# Associations of perceived neighborhood factors and Alzheimer's disease polygenic score with cognition: Evidence from the Health and Retirement Study

**Erin B. Ware**[1,2☯*], **Peiyao Zhu**[3☯], **Grace Noppert**[1], **Mingzhou Fu**[3], **Mikayla Benbow**[3], **Lindsay C. Kobayashi**[1,3], **Lindsay H. Ryan**[1], **Kelly M. Bakulski**[3]

**1** Survey Research Center, Institute for Social Research Center, University of Michigan, Ann Arbor, Michigan, United States of America, **2** Population Studies Center, Institute for Social Research Center, University of Michigan, Ann Arbor, Michigan, United States of America, **3** School of Public Health, University of Michigan, Ann Arbor, Michigan, United States of America

☯ These authors contributed equally to this work.

\* ebakshis@umich.edu

## Abstract

### Background

We examined the relationships between neighborhood characteristics, cumulative genetic risk for Alzheimer's disease (polygenic scores for Alzheimer's disease), and cognitive function using data from the Health and Retirement Study (2008–2020, age > 50).

### Methods

Baseline perceived neighborhood characteristics were combined into a subjective neighborhood disadvantage index. Cognitive function was assessed at baseline and measured biennially over a 10-year follow-up period. Analyses were stratified by genetic ancestry. Cox proportional hazard models analyzed associations between neighborhood characteristics, Alzheimer's disease polygenic scores, and their interactions on cognitive impairment.

### Results

In the European ancestries sample, a one standard deviation higher score on the subjective neighborhood disadvantage index was associated with a higher hazard of any cognitive impairment (HR:1.09; CI:1.03–1.15), cognitive impairment without dementia (HR:1.08; CI:1.03–1.14), and dementia (HR:1.13; CI:1.03–1.24). Similarly, a one standard deviation increase in Alzheimer's disease polygenic score was associated with a higher risk of cognitive impairment (HR:1.10; CI:1.05–1.16) and cognitive impairment without dementia (HR:1.10; CI:1.05–1.16) but not dementia (HR:1.05;

**Data availability statement:** This analysis uses data from the Health and Retirement Study, polygenic scores, leave-behind questionnaire, Imputation of Cognitive Functioning Measures, and core study, sponsored by the National Institute on Aging (grant number NIA U01AG009740) and conducted by the University of Michigan. The data that support the findings of this study are publicly available at https://hrs.isr.umich.edu/.

**Funding:** This work was supported by the National Institute on Aging (NIA): EBW (Erin B. Ware): R01 AG055406. KMB (Kelly M. Bakulski): R01 AG067592, P30 AG072931. GAN (Grace A. Noppert): R01 AG075719. LCK (Lindsay C. Kobayashi): R01 AG070953.

**Competing interests:** The authors have declared that no competing interests exist.

**Abbreviations:** CI, confidence interval; SD, standard deviation; SNPs, single-nucleotide polymorphisms; GWAS, genome wide association study; BMI, body mass index; APOE, Apolipoprotein E.

CI:0.96–1.16). No significant interactions were found. Evidence in African ancestries were directionally similar but imprecise and inconclusive due to limited precision and cross-ancestry polygenic score transferability. Subjective neighborhood disadvantage index and Alzheimer's disease polygenic score were independently associated with incident cognitive impairment.

## Conclusions

Preventing dementia by addressing modifiable risk factors is essential.

## Introduction

Dementia describes a group of progressive neurological disorders that severely affect individuals' memory, thinking, and social abilities [1]. The most common type of dementia is Alzheimer's disease, with pathology present in 60–80% of cases [2]. By the year 2050, an estimated 12.7 million Americans aged 65 years and older will be living with Alzheimer's disease [3]. In 2022, the estimated medical cost associated with Alzheimer's disease or other dementias amounted to $321 billion in the United States, with spending expected to exceed $1 trillion in 2050 [4], leading to a significant financial burden at both the family and national levels. Cognitive impairment non-dementia (CIND) is an intermediate stage between normal cognition and dementia [5–7]. Identifying modifiable risk factors for cognitive impairment is critical because addressing these factors during intermediate stages of cognitive decline can enhance the effectiveness of public health interventions, potentially reducing the progression to dementia and its related societal burden. In addition, understanding how these factors interact with individual susceptibilities can provide deeper insight into the pathways influencing cognitive decline and help inform strategies to reduce risk across diverse populations.

The neighborhood environment is a potential modifiable risk factor for cognitive health. Evidence suggests that neighborhood environment has been linked to risk of cognitive impairment and dementia [8,9], with both objective and subjective assessments of neighborhoods offering distinct insights into the association between neighborhoods and cognition [10]. Objective characteristics of neighborhoods, such as crime rates, access to amenities and neighborhood socioeconomic status, are often used to assess both neighborhood quality and the accessibly of resources in public health studies [10,11]. For example, in the French Three City Cohort, a population-based prospective study, women who lived in a deprived neighborhood—defined as having a low median income—had an increased risk of developing dementia and Alzheimer's disease relative to those living in non-deprived neighborhoods [8]. In a cross-sectional study of 21,008 adults aged 65 years and older from Hong Kong, more walkable neighborhoods, measured by proximity to frequently-used amenities, were associated with 3.5% lower prevalence of dementia relative to the least walkable neighborhoods [12]. Subjective characteristics of neighborhoods—residents' perceptions of their neighborhood environment—can be equally or more important in determining health outcome,

as they directly reflect the mental and emotional responses to the environment which are not always captured by objective measures [10,13,14]. Feelings of safety, cleanliness, and social support can influence leisure-time physical activity and chronic stress [15,16]. A sense of neighborhood belonging, and the friendliness of neighbors can influence social connectedness and feelings of loneliness [17,18]. These factors collectively contribute to social engagement and overall mental health and are critical in maintaining cognitive function [19–21].

Genetics also play a major role in the risk of developing cognitive impairment and dementia [22,23]. Variation in the Apolipoprotein E (*APOE*) gene is a major genetic risk factor for Alzheimer's disease [24]. There are three major gene variants of the *APOE* gene, the ε2, ε3, and ε4 alleles [24]. At later ages, those with one copy of the *APOE*-ε4 allele have approximately three times increased risk of Alzheimer's disease, and those with two copies of ε4 have 8–14 times increased risk, compared to the ε3/ε3 genotype [22]. However, genetics alone are neither sufficient nor necessary to cause dementia and it is important to understand the interplay between genetics and other factors on the risk of developing cognitive impairment.

Both genetic and environmental factors together contribute to the risk of cognitive impairment [25]. Gene-by-environment interactions occur when the effect of an environment is dependent on the genotype of the individual [26]. For example, in a longitudinal study of primary care patients aged 75 years or older, those with both the *APOE*-ε4 allele and low physical activity (defined as less than one physical activity per week) had a higher conversion rate (21.5%) to dementia and Alzheimer's disease, as well as a higher relative risk (relative risk: 3.02, 95% CI: 2.07, 4.42) of dementia and Alzheimer's disease, compared to those who had only one of the risk factors present [27]. Investigating gene-by-environment interactions, involving neighborhoods and genetic risk factors for dementia, enhances our understanding of the manifestation of diseases and significantly impact precision medicine by enabling targeted risk prevention for individuals highly susceptible to specific environmental effects [28,29].

In this study, we conducted a Cox proportional hazards analysis using data from participants of European and African genetic ancestries in the United States Health and Retirement Study. Our goal was to examine the associations between perceived neighborhood characteristics and incident cognitive impairment. We categorized cognitive impairment as 1) any cognitive impairment (CIND or dementia), 2) CIND, or 3) dementia. We stratified analyses by genetic ancestry group since the polygenic scores are on different scales due to the genetic architecture of the samples. Additionally, we assessed whether the association between perceived neighborhood characteristics and cognitive impairment was modified by Alzheimer's disease polygenic score.

## Methods

### Health and Retirement Study

The Health and Retirement Study is a nationally representative, longitudinal panel study of people in the United States 50 years and older [30,31]. The Health and Retirement Study is funded by the National Institute on Aging (NIA U01AG009740) and the Social Security Administration. The Institute for Social Research at the University of Michigan conducts the Health and Retirement study. Data are collected through in-person and telephone interviews in waves occurring every two years since the study's start in 1992. To maintain its nationally representative status, the sample of participants is replenished every six years with new cohorts. Data were accessed between 1 October 2017 and 1 June 2025 for research purposes. The authors had no access to information that could identify individual participants during or after data collection.

### Neighborhood measures

Half of the Health and Retirement Study participants in 2008 were randomly selected to answer questions on their neighborhoods, while the other half were asked these questions during the 2010 wave [31]. In our analysis, we established a baseline for participants based in the wave in which they were interviewed for the neighborhood questions (either 2008 or 2010).

Participants were asked to rate aspects of their neighborhood as a part of a leave-behind questionnaire. Neighborhoods were defined as a participant's 'local area' or everywhere 'within a 20-minute walk or about a mile of their house.' Participant-evaluated neighborhood characteristics include: *1) There is no problem with vandalism and graffiti in this area/Vandalism and graffiti are a big problem in this area, 2) Most people in this area can be trusted/Most people in this area can't be trusted, 3) People feel safe walking alone in this area after dark/People would be afraid to walk alone in this area after dark, 4) Most people in this area are friendly/Most people in this area are unfriendly, 5) This area is kept very clean/This area is always full of rubbish and litter, 6) There are no vacant houses or storefronts in this area/There are many vacant or deserted houses or storefronts in this area. 7) People feel that they are part of/don't belong in this area.* Participants were instructed to choose a value of one to seven for each of the questions, with one corresponding to the first response scale and seven corresponding to the latter response scale of each question. The responses to each neighborhood question were standardized (mean=0, standard deviation=1). Given correlations among responses to these seven questions, we created a subjective neighborhood disadvantage index using principal components analysis to identify factors representing the most variation in the neighborhood question responses [32]. Higher subjective neighborhood disadvantage indexes indicate more disadvantaged neighborhoods. Secondary analyses considered a dichotomous version of the subjective neighborhood disadvantage index, with scores of 0 or below indicating the least disadvantaged neighborhoods and scores above 0 indicating the most disadvantaged neighborhoods. Additionally, we conducted supplemental analyses using each neighborhood characteristic as a separate exposure.

## Cognitive status

The Health and Retirement Study collects information on participants' cognition in every data collection wave. We assessed participants' cognitive function at baseline and followed up with up to five waves of cognitive assessments over the subsequent ten years, from 2010 to 2018 for those with a 2008 baseline, and from 2012 to 2020 for those with a 2010 baseline. The Health and Retirement Study cognitive assessments include ten word immediate and delayed recall, serial 7s subtraction, counting backwards, object naming and recall of date and president and vice-president. These tests assess memory, working memory, attention and processing speed, language, and orientation and are administered via telephone [33].

We used the Health and Retirement Study Imputation of Cognitive Functioning Measures data which provides cleaned and imputed cognitive function assessment scores [34]. Proxy respondents were excluded from our study sample at each wave. To categorize cognitive status in the Health and Retirement Study, we used the Langa-Weir approach, which uses a composite score of 0–27 from the cognitive assessments available in the Health and Retirement Study (excluding orientation and naming objects). Individuals with scores from 0–6 were categorized as having dementia, 7–11 as CIND, and 12–27 as normal cognition [35].

## Polygenic score for Alzheimer's disease

The Health and Retirement Study collected saliva samples in 2006, 2008, 2010, or 2012 for DNA processing. In 2006, the mouthwash collection method was used, then from 2008 on, the Oragene DNA Collection Kit (OG250) was used. Approximately 2.4 million single nucleotide polymorphisms (SNPs) were measured using the Ilumina HumanOmni2.5 BeadChip array (HumanOmni2.5-4v1, HumanOmni2.5-8v1). The University of Washington Genetics Coordinating Center performed genotyping quality control. SNPs were aligned to genome build 37/hg 19, phased using SHAPEIT2, and genetic principal components were calculated with HapMap controls [36,37]. Imputation was conducted by the University of Washington Genetics Coordinating Center using IMPUTE2, to impute approximately 21 million SNPs to the 1000 Genomes Project cosmopolitan reference panel phase 3 version 5 [38]. Genetic data for the Health and Retirement Study is available through application to the National Institute on Aging Genetics of Alzheimer's Disease Data Storage Site (https://dss.niag-ads.org/datasets/ng00153/).

The Health and Retirement Study used both self-identified race/ethnicity and ancestral genetic similarity to ascertain the African and European ancestries analytic samples. To account for population stratification within the two broad

ancestries groups, local genetic principal components were calculated within each group and used in downstream analyses, provided by the Health and Retirement Study [38,39].

A polygenic score provides a quantitative summary measure of the genetic predisposition to express a the given trait [40]. The Health and Retirement Study provides polygenic scores from the International Genomics of Alzheimer's Project (IGAP) using SNPs associated with Alzheimer's disease at two genome-wide association p-value thresholds (pT): pT = 0.01 and pT = 1.0 [41]. In this study, we used a p-value threshold of pT = 0.01 and did not include linkage disequilibrium in the scores (i.e., no pruning algorithms). The *APOE* gene region (start: 43.38 megabases, stop: 45.43 megabases, build hg37) was removed from the polygenetic score because including it through weighted variants was insufficient to capture the substantial risk attributed to this region [42]. Therefore, we treated the *APOE* region as a separate covariate to better account for its independent effect on dementia risk. Our primary analysis used z-score standardized polygenic scores (mean = 0, standard deviation = 1) within each genetic ancestry group. In subsequent analyses, we dichotomized the polygenic score for Alzheimer's disease without the *APOE* region at the 75th percentile to represent those in the highest quartile of genetic risk for Alzheimer's disease [43].

## Covariate measures

We selected covariates that could be confounders of the association between neighborhood characteristics and cognitive status, based on prior evidence [44,45]. Covariates were assessed at the analytical baseline (2008 or 2010) and were: age (continuous, in years), sex (male; female) highest educational attainment (above high school/GED; high school/GED; less than high school/GED), subjective social status (self-reported standing on a ladder representing "society", continuous range 1 [low] to 10 [high]) [46], and poverty status, defined as whether household income was below the US Census poverty threshold (yes; no). Additionally, we included an indicator variable for baseline wave (wave 2008; wave 2010) to account for potential cohort differences.

In sensitivity analyses, we considered the additional potential confounders of smoking (current, former, never), alcohol consumption (number of drinks/day when drinks), history of type 2 diabetes (yes, no), depressive symptoms (continuous), body mass index (BMI; continuous, in $kg/m^2$), self-rated hearing (excellent, very good, good, fair, poor), self-rated vision (excellent, very good, good, fair, poor, blind), brain-related conditions (yes, no to a report of stroke and/or psychiatric problems) and number of physician-diagnosed chronic medical conditions (0, 1–2, 3 or more of the following: high blood pressure, diabetes, cancer, lung disease, heart disease, and arthritis), all measured at the analytical baseline (2008 or 2010). We considered these covariates in a sensitivity analysis rather than the main analysis, as although they have been associated with cognitive status [47–50], they are time-varying and could be influenced by neighborhood factors. Therefore, these variables may lie on the causal pathway rather than serve as confounders.

## Sample selection

Eligible Health and Retirement Study participants were those who were interviewed in the 2008 and 2010 waves with complete neighborhood data (n = 14,562). Then, we excluded participants with incomplete genetic (n = 5,220), cognition (n = 717), and covariate data (n = 717) (Fig 1 and S1 Table). Additionally, we excluded participants with dementia at baseline (N = 148) from analyses where dementia was the outcome. For analyses of any cognitive impairment (CIND and dementia), participants with CIND and dementia at baseline (N = 1,082) were excluded. In CIND-specific analyses, we further excluded participants who developed dementia during follow-up without prior records of CIND (N = 80) (Fig 1 and S1 Table).

## Statistical analysis

All the analyses were stratified by genetic ancestry (European ancestries and African ancestries). In bivariate analyses, we described the distributions of variables by cognitive functions using t-test (mean, SD) for continuous variables and

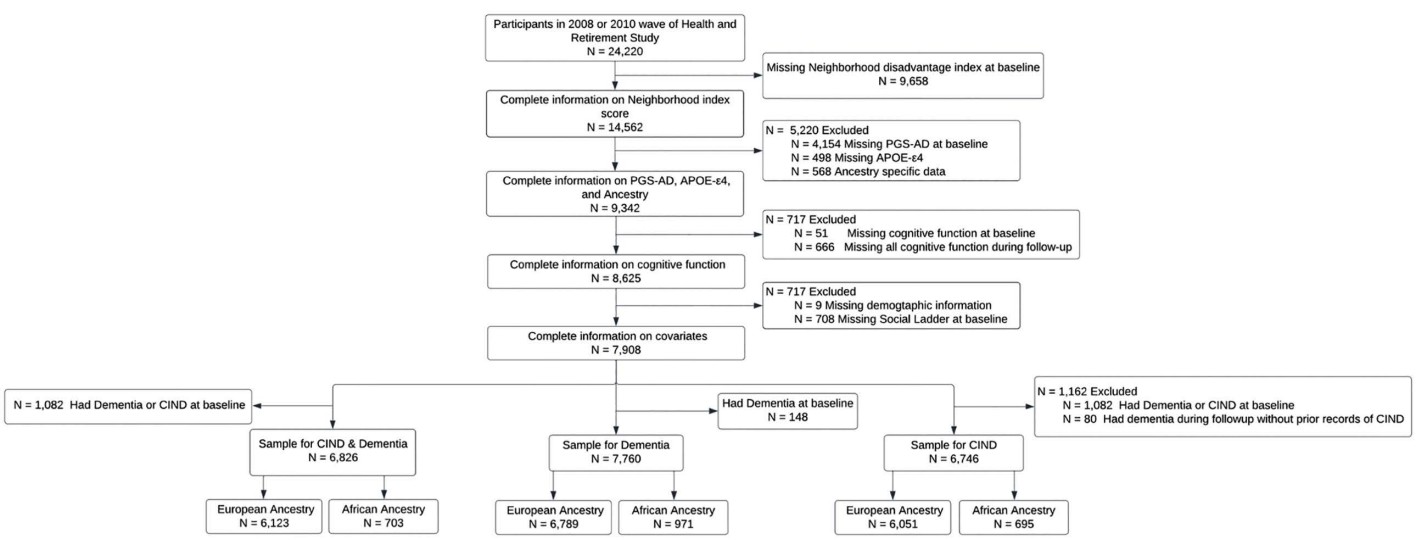

**Fig 1. Sample inclusion and exclusion flow chart for the Health and Retirement Study 2018 and 2010 wave.**

chi-square test (count, %) for categorical variables. We similarly tested for differences between the included and excluded sample to note selection bias in the included sample.

We used Cox proportional hazard modeling to examine the associations between the subjective neighborhood disadvantage index and incident cognitive impairment, with effect modification by Alzheimer's disease polygenic score. In model 1, we include adjustments for sociodemographic factors (baseline age, sex, education, poverty status, subjective social status), *APOE*-ε4 carrier status, and baseline year indicator. Model 2 included additional adjustment for Alzheimer's disease polygenic score and genetic principal components. Model 3 further tested the interaction between the subjective neighborhood disadvantage index and Alzheimer's disease polygenic score. The subjective neighborhood disadvantage index, covariates, and supplementary variables were measured at baseline, with cognitive function assessed at baseline and across five follow-up waves over 10 years.

The following fully adjusted model was fitted to our dataset:

$$\lambda\left(t|X\right) = \lambda_0(t)exp(\beta_1 Neighborhood + \beta_2 PGS\_AD + \beta_3(Neighborhood \times PGS\_AD) + \beta_4 Covariates) \tag{1}$$

where $\lambda\left(t|X\right)$ is the hazard function at time *t* given covariates *X*, and $\lambda_0(t)$ is the baseline hazard at *t*ime t. We performed three separate comparisons using this model specification stratified by ancestries: 1) any incident cognitive impairment (CIND or dementia) as the outcome compared to normal cognition, 2) incident CIND as the outcome compared to normal cognition and 3) incident dementia as the outcome compared to no dementia, where "no dementia" includes both normal cognition and CIND. This approach maintains sample size by keeping individuals who developed CIND during follow-up, who have a higher risk of progressing to dementia, in the reference group until dementia onset. We considered P values <0.05 as indicating statistical significance.

We conducted analyses in SAS (9.4 version) and R statistical software (4.3.3 version).

## Relative excess risk due to interaction

Additive interaction, a more relevant public health measure than multiplicative interaction, is often emphasized in epidemiology because it reflects the absolute difference in risk associated with combined exposures [51,52]. Therefore, we

calculated both multiplicative and additive interaction in the study. To assess additive interaction, we evaluated Cox proportional hazard models using dichotomized subjective neighborhood disadvantage index and Alzheimer's disease polygenic score. We then calculated relative excess risk due to interaction (RERI), to further elucidate the additive interaction between subjective neighborhood disadvantage index and Alzheimer's disease polygenic score on cognitive impairment, due to its outstanding performance in Cox proportional hazard models [53]. The RERI was calculated using the "*epiR*" package [54] in R statistical software (4.3.3 version). A positive RERI indicates a positive interaction or more than additivity, zero indicates no interaction, and a negative value indicates negative interaction or less than additivity [55].

### Sensitivity analysis

In our first sensitivity models, we included a third set of variables as covariates—smoking status, alcohol consumption, BMI, diabetes, self-rated hearing, self-rated vision, brain-related conditions, number of chronic conditions, and depression—which are time-varying factors that may lie on the causal pathway rather than act as confounders. Additionally, to determine if any individual neighborhood factor (perceived safety, trust, friendly, cleanness, vandalism, vacant, belonging—each represented by one of the seven questions in the neighborhood measures) drives the observed associations, we evaluated each individual neighborhood factor as exposures in modeling.

### Ethics

This secondary analysis was approved by the University of Michigan Institutional Review Board (HUM00128220). Participants provided written informed consent. Research practices at the University of Michigan are guided by the ethical principles outlined in the Belmont Report, which include respect for persons, beneficence, and justice.

## Results

### Descriptive analysis

In our study sample, participants who developed cognitive impairment tended to be older, female, have higher subjective neighborhood disadvantage indexes (indicating lower perceived safety, friendliness and environment), more likely to have lower education levels, and reported lower subjective social status (Table 1). The sample distribution, stratified by genetic ancestries, included 703 participants of African ancestries and 6,123 of European ancestries (N = 6,826) for cognitive impairment sample; 695 of African ancestries and 6,051 of European ancestries (N = 6,746) for CIND sample; and 971 of African ancestries and 6,789 of European ancestries (N = 7,760) for dementia sample (Fig 1 and S2 Table). Additionally, individuals' responses to the seven neighborhood characteristics were correlated (Fig 2).

### European ancestries sample

**Multivariable analysis.** In the European ancestries sample, each standard deviation increase in the subjective neighborhood disadvantage index was associated with an 9% higher risk of developing any cognitive impairment (95% CI: 1.03, 1.15), an 8% higher risk of CIND (95% CI: 1.02–1.14), and a 13% higher risk of dementia (95% CI: 1.02–1.24) in the fully adjusted model (Model 3 in Table 2). In the fully adjusted models, each standard deviation increase in the Alzheimer's disease polygenic score was associated with a 10% increased risk of any cognitive impairment and CIND (95% CI: 1.05–1.16). The Alzheimer's disease polygenic score was not significantly associated with dementia. No significant multiplicative interaction was found between the subjective neighborhood disadvantage index and Alzheimer's disease polygenic score for any of the outcomes in the European ancestries sample (Table 2).

**RERI analysis.** In models using dichotomous exposures (subjective neighborhood disadvantage index and Alzheimer's disease polygenic score), individuals living in the more disadvantaged neighborhoods had a 21% higher risk of any cognitive impairment (95% CI: 1.08–1.36), a 20% higher risk of CIND (95% CI: 1.06–1.35), and a 34%

**Table 1. Sample distribution of neighborhood disadvantage index, cognitive function, genetic characteristics, sociodemographic, behavioral, and chronic conditions by cognitive impairment (CIND and dementia), CIND, and dementia in the Health and Retirement Study, wave 2008-2010.**

| Main Variables | Cognitive Impairment[c] | | | | CIND[c] | | | | Dementia[c] | | | |
|---|---|---|---|---|---|---|---|---|---|---|---|---|
| | Overall[a] | Normal[a] | Cognitive Impairment[a] | p-value[b] | Overall[a] | Normal[a] | CIND[a] | p-value[b] | Overall[a] | Non-dementia[a] | Dementia[a] | p-value[b] |
| | N = 6,826 | N = 4,754 | N = 2,072 | | N = 6,746 | N = 4,754 | N = 1,992 | | N = 7,760 | N = 7,142 | N = 618 | |
| **Neighborhood disadvantage index** | −0.13 (0.90) | −0.16 (0.86) | −0.05 (0.97) | <0.001 | −0.13 (0.90) | −0.16 (0.86) | −0.06 (0.97) | <0.001 | −0.10 (0.93) | −0.11 (0.92) | 0.05 (1.03) | <0.001 |
| **Neighborhood disadvantage index (Binary)** | | | | <0.001 | | | | <0.001 | | | | <0.001 |
| The least disadvantaged neighborhoods (<=0) | 4,537 (66%) | 3,234 (68%) | 1,303 (63%) | | 4,488 (67%) | 3,234 (68%) | 1,254 (63%) | | 5,070 (65%) | 4,716 (66%) | 354 (57%) | |
| The most disadvantaged neighborhoods (>0) | 2,289 (34%) | 1,520 (32%) | 769 (37%) | | 2,258 (33%) | 1,520 (32%) | 738 (37%) | | 2,690 (35%) | 2,426 (34%) | 264 (43%) | |
| **Baseline wave** | | | | <0.001 | | | | <0.001 | | | | <0.001 |
| Wave 1 (2008) | 3,008 (44%) | 2,023 (43%) | 985 (48%) | | 2,971 (44%) | 2,023 (43%) | 948 (48%) | | 3,436 (44%) | 3,120 (44%) | 316 (51%) | |
| Wave 2 (2010) | 3,818 (56%) | 2,731 (57%) | 1,087 (52%) | | 3,775 (56%) | 2,731 (57%) | 1,044 (52%) | | 4,324 (56%) | 4,022 (56%) | 302 (49%) | |
| **Ancestry** | | | | <0.001 | | | | <0.001 | | | | <0.001 |
| European ancestry | 6,123 (90%) | 4,356 (92%) | 1,767 (85%) | | 6,051 (90%) | 4,356 (92%) | 1,695 (85%) | | 6,789 (87%) | 6,337 (89%) | 452 (73%) | |
| African ancestry | 703 (10%) | 398 (8.4%) | 305 (15%) | | 695 (10%) | 398 (8.4%) | 297 (15%) | | 971 (13%) | 805 (11%) | 166 (27%) | |
| **PGS-AD** | | | | | | | | | | | | |
| European ancestry | −0.08 (0.98) | −0.12 (0.98) | 0.00 (0.96) | — | −0.09 (0.98) | −0.12 (0.98) | −0.01 (0.96) | — | −0.08 (0.98) | −0.08 (0.98) | −0.05 (0.94) | — |
| African ancestry | 0.06 (0.93) | −0.04 (0.94) | 0.19 (0.90) | — | 0.06 (0.93) | −0.04 (0.94) | 0.18 (0.90) | — | 0.06 (0.91) | 0.05 (0.92) | 0.10 (0.83) | — |
| **PGS-AD (Binary)** | | | | | | | | | | | | |
| European ancestry | | | | — | | | | — | | | | — |
| Below 75% | 4,769 (78%) | 3,429 (79%) | 1,340 (76%) | | 4,719 (78%) | 3,429 (79%) | 1,290 (76%) | | 5,267 (78%) | 4,930 (78%) | 337 (75%) | |
| Above 75% | 1,354 (22%) | 927 (21%) | 427 (24%) | | 1,332 (22%) | 927 (21%) | 405 (24%) | | 1,522 (22%) | 1,407 (22%) | 115 (25%) | |
| African ancestry | | | | — | | | | — | | | | — |
| Below 75% | 534 (76%) | 315 (79%) | 219 (72%) | | 531 (76%) | 315 (79%) | 216 (73%) | | 741 (76%) | 613 (76%) | 128 (77%) | |
| Above 75% | 169 (24%) | 83 (21%) | 86 (28%) | | 164 (24%) | 83 (21%) | 81 (27%) | | 230 (24%) | 192 (24%) | 38 (23%) | |
| **APOE E4 status** | | | | <0.001 | | | | <0.001 | | | | <0.001 |
| Any copies of e4 | 1,807 (26%) | 1,200 (25%) | 607 (29%) | | 1,783 (26%) | 1,200 (25%) | 583 (29%) | | 2,094 (27%) | 1,860 (26%) | 234 (38%) | |
| No copies of e4 | 5,019 (74%) | 3,554 (75%) | 1,465 (71%) | | 4,963 (74%) | 3,554 (75%) | 1,409 (71%) | | 5,666 (73%) | 5,282 (74%) | 384 (62%) | |
| **Sex** | | | | 0.015 | | | | 0.018 | | | | 0.476 |
| Male | 2,674 (39%) | 1,818 (38%) | 856 (41%) | | 2,641 (39%) | 1,818 (38%) | 823 (41%) | | 3,093 (40%) | 2,855 (40%) | 238 (39%) | |
| Female | 4,152 (61%) | 2,936 (62%) | 1,216 (59%) | | 4,105 (61%) | 2,936 (62%) | 1,169 (59%) | | 4,667 (60%) | 4,287 (60%) | 380 (61%) | |
| **Age** | 66.61 (10.06) | 64.74 (9.54) | 70.91 (9.90) | <0.001 | 66.54 (10.04) | 64.74 (9.54) | 70.83 (9.88) | <0.001 | 67.29 (10.28) | 66.68 (10.09) | 74.34 (9.76) | <0.001 |
| **Education Level** | | | | <0.001 | | | | <0.001 | | | | <0.001 |
| >High School/ GED | 2,362 (35%) | 1,888 (40%) | 474 (23%) | | 2,336 (35%) | 1,888 (40%) | 448 (22%) | | 2,489 (32%) | 2,394 (34%) | 95 (15%) | |
| High School/ GED | 3,921 (57%) | 2,643 (56%) | 1,278 (62%) | | 3,876 (57%) | 2,643 (56%) | 1,233 (62%) | | 4,425 (57%) | 4,103 (57%) | 322 (52%) | |
| <High School/ GED | 543 (8.0%) | 223 (4.7%) | 320 (15%) | | 534 (7.9%) | 223 (4.7%) | 311 (16%) | | 846 (11%) | 645 (9.0%) | 201 (33%) | |
| **Poverty Status** | | | | <0.001 | | | | <0.001 | | | | <0.001 |
| Above Poverty threshold | 6,518 (95%) | 4,574 (96%) | 1,944 (94%) | | 6,441 (95%) | 4,574 (96%) | 1,867 (94%) | | 7,320 (94%) | 6,780 (95%) | 540 (87%) | |
| Below Poverty threshold | 308 (4.5%) | 180 (3.8%) | 128 (6.2%) | | 305 (4.5%) | 180 (3.8%) | 125 (6.3%) | | 440 (5.7%) | 362 (5.1%) | 78 (13%) | |

*(Continued)*

**Table 1.** (Continued)

| Main Variables | Cognitive Impairment[c] | | | | CIND[c] | | | | Dementia[c] | | | |
|---|---|---|---|---|---|---|---|---|---|---|---|---|
| | Overall[a] | Normal[a] | Cognitive Impairment[a] | p-value[b] | Overall[a] | Normal[a] | CIND[a] | p-value[b] | Overall[a] | Non-dementia[a] | Dementia[a] | p-value[b] |
| | N = 6,826 | N = 4,754 | N = 2,072 | | N = 6,746 | N = 4,754 | N = 1,992 | | N = 7,760 | N = 7,142 | N = 618 | |
| **Social Ladder** | 6.55 (1.66) | 6.63 (1.63) | 6.36 (1.70) | <0.001 | 6.55 (1.66) | 6.63 (1.63) | 6.35 (1.71) | <0.001 | 6.51 (1.68) | 6.52 (1.67) | 6.37 (1.81) | 0.049 |
| **Sensitivity Variables** | Overall* | Normal | CIND | p-value | Overall* | Normal | CIND | p-value | Overall* | Non-dementia | Dementia | p-value |
| **Smoking status** | | | | 0.019 | | | | 0.02 | | | | 0.094 |
| Current Smoker | 819 (12%) | 569 (12%) | 250 (12%) | | 813 (12%) | 569 (12%) | 244 (12%) | | 946 (12%) | 883 (12%) | 63 (10%) | |
| Former Smoker | 2,917 (43%) | 1,981 (42%) | 936 (45%) | | 2,877 (43%) | 1,981 (42%) | 896 (45%) | | 3,358 (44%) | 3,068 (43%) | 290 (47%) | |
| Never Smoke | 3,050 (45%) | 2,178 (46%) | 872 (42%) | | 3,018 (45%) | 2,178 (46%) | 840 (42%) | | 3,411 (44%) | 3,150 (44%) | 261 (43%) | |
| **BMI** | 28.67 (6.00) | 28.74 (6.01) | 28.50 (5.99) | 0.136 | 28.68 (6.00) | 28.74 (6.01) | 28.55 (5.97) | 0.248 | 28.60 (6.04) | 28.69 (6.05) | 27.66 (5.79) | <0.001 |
| **Drinking (# drinks/day when drinks)** | 0.81 (1.37) | 0.86 (1.35) | 0.69 (1.41) | <0.001 | 0.81 (1.37) | 0.86 (1.35) | 0.69 (1.42) | <0.001 | 0.78 (1.38) | 0.80 (1.38) | 0.55 (1.42) | <0.001 |
| **Ever have Diabetes** | | | | <0.001 | | | | <0.001 | | | | 0.037 |
| Yes | 1,129 (17%) | 711 (15%) | 418 (20%) | | 1,119 (17%) | 711 (15%) | 408 (20%) | | 1,369 (18%) | 1,241 (17%) | 128 (21%) | |
| No | 5,697 (83%) | 4,043 (85%) | 1,654 (80%) | | 5,627 (83%) | 4,043 (85%) | 1,584 (80%) | | 6,391 (82%) | 5,901 (83%) | 490 (79%) | |
| **Brain Condition** | | | | <0.001 | | | | <0.001 | | | | <0.001 |
| No | 5,516 (81%) | 3,908 (82%) | 1,608 (78%) | | 5,456 (81%) | 3,908 (82%) | 1,548 (78%) | | 6,211 (80%) | 5,751 (81%) | 460 (74%) | |
| Yes | 1,310 (19%) | 846 (18%) | 464 (22%) | | 1,290 (19%) | 846 (18%) | 444 (22%) | | 1,549 (20%) | 1,391 (19%) | 158 (26%) | |
| **Chronic Condition** | | | | <0.001 | | | | <0.001 | | | | <0.001 |
| None | 1,194 (17%) | 943 (20%) | 251 (12%) | | 1,181 (18%) | 943 (20%) | 238 (12%) | | 1,303 (17%) | 1,228 (17%) | 75 (12%) | |
| 1 - 2 | 3,899 (57%) | 2,771 (58%) | 1,128 (54%) | | 3,858 (57%) | 2,771 (58%) | 1,087 (55%) | | 4,349 (56%) | 4,018 (56%) | 331 (54%) | |
| >= 3 | 1,733 (25%) | 1,040 (22%) | 693 (33%) | | 1,707 (25%) | 1,040 (22%) | 667 (33%) | | 2,108 (27%) | 1,896 (27%) | 212 (34%) | |
| **Eyesight** | | | | <0.001 | | | | <0.001 | | | | <0.001 |
| Excellent | 729 (11%) | 561 (12%) | 168 (8.1%) | | 721 (11%) | 561 (12%) | 160 (8.0%) | | 793 (10%) | 746 (10%) | 47 (7.6%) | |
| Very Good | 2,214 (32%) | 1,661 (35%) | 553 (27%) | | 2,191 (32%) | 1,661 (35%) | 530 (27%) | | 2,406 (31%) | 2,279 (32%) | 127 (21%) | |
| Good | 2,886 (42%) | 1,979 (42%) | 907 (44%) | | 2,853 (42%) | 1,979 (42%) | 874 (44%) | | 3,295 (42%) | 3,032 (42%) | 263 (43%) | |
| Fair | 793 (12%) | 445 (9.4%) | 348 (17%) | | 781 (12%) | 445 (9.4%) | 336 (17%) | | 983 (13%) | 861 (12%) | 122 (20%) | |
| Poor | 193 (2.8%) | 103 (2.2%) | 90 (4.3%) | | 190 (2.8%) | 103 (2.2%) | 87 (4.4%) | | 268 (3.5%) | 211 (3.0%) | 57 (9.2%) | |
| Blind | 8 (0.1%) | 4 (<0.1%) | 4 (0.2%) | | 7 (0.1%) | 4 (<0.1%) | 3 (0.2%) | | 12 (0.2%) | 10 (0.1%) | 2 (0.3%) | |
| **Hearing** | | | | <0.001 | | | | <0.001 | | | | <0.001 |
| Excellent | 1,278 (19%) | 965 (20%) | 313 (15%) | | 1,269 (19%) | 965 (20%) | 304 (15%) | | 1,419 (18%) | 1,331 (19%) | 88 (14%) | |
| Very Good | 2,127 (31%) | 1,603 (34%) | 524 (25%) | | 2,105 (31%) | 1,603 (34%) | 502 (25%) | | 2,315 (30%) | 2,177 (30%) | 138 (22%) | |
| Good | 2,251 (33%) | 1,501 (32%) | 750 (36%) | | 2,223 (33%) | 1,501 (32%) | 722 (36%) | | 2,596 (33%) | 2,367 (33%) | 229 (37%) | |
| Fair | 911 (13%) | 543 (11%) | 368 (18%) | | 892 (13%) | 543 (11%) | 349 (18%) | | 1,098 (14%) | 975 (14%) | 123 (20%) | |
| Poor | 257 (3.8%) | 141 (3.0%) | 116 (5.6%) | | 255 (3.8%) | 141 (3.0%) | 114 (5.7%) | | 330 (4.3%) | 291 (4.1%) | 39 (6.3%) | |
| **Depression** | 1.08 (1.71) | 0.96 (1.62) | 1.35 (1.86) | <0.001 | 1.08 (1.71) | 0.96 (1.62) | 1.35 (1.88) | <0.001 | 1.15 (1.76) | 1.11 (1.74) | 1.60 (1.94) | <0.001 |

CIND: Cognitive impairment non-dementia, PGS-AD: Polygenic score – Alzheimer's disease.

[a] n (%); Mean (SD).

[b] Pearson's Chi-squared test; Welch Two Sample t-test.

[c] Average follow-up duration: Cognitive Impairment=7.41 years; Cognitive Impairment, No Dementia (CIND) = 7.45 years; Dementia=8.02 years.

* Sample size after excluding all missing values, including sensitivity analysis covariates: for Cognitive Impairment analysis: 6,727 , CIND: 6,650, Dementia: 7,645.

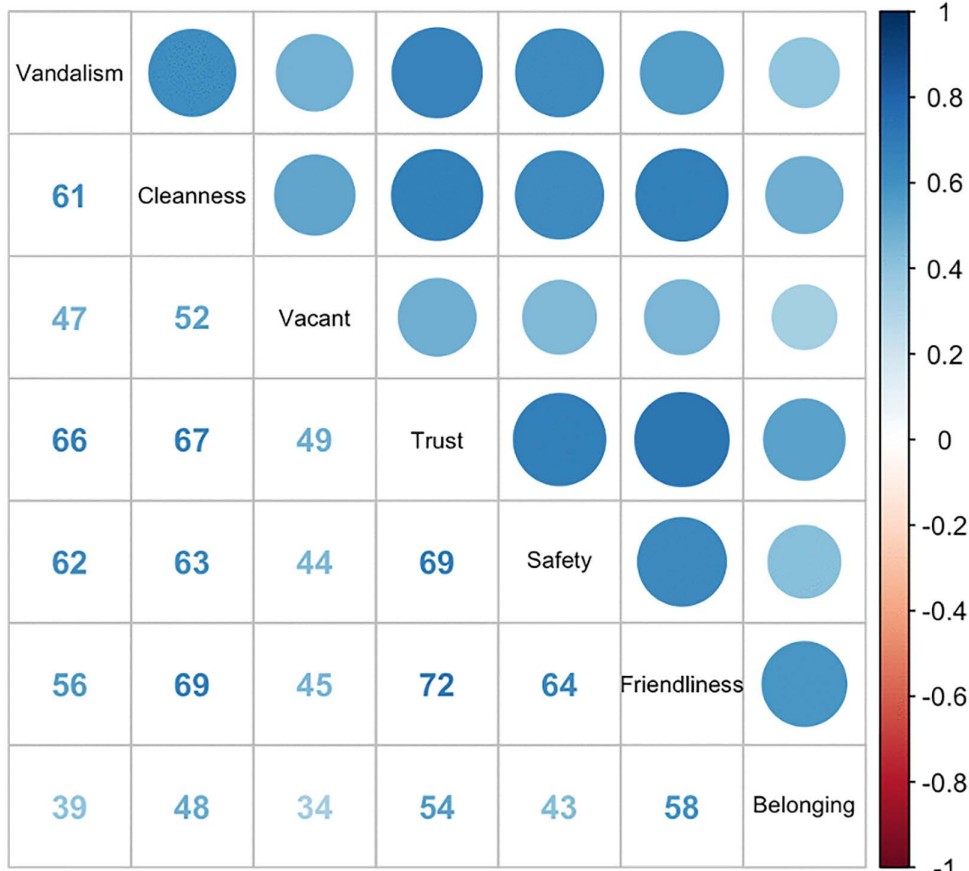

**Fig 2. Correlations between neighborhood characteristics in the Health and Retirement Study 2018 and 2010 wave.**

higher risk of dementia (95% CI: 1.07–1.68) compared to those in the least disadvantaged neighborhoods (S3 Table). Participants with an Alzheimer's disease polygenic score in the top 25% of the sample had a 12% higher risk of developing any cognitive impairment (95% CI: 1.01–1.25). However, no significant associations were observed between the dichotomous Alzheimer's disease polygenic score and the risk of CIND (HR: 1.13; 95% CI: 0.99–1.30) or dementia (HR: 1.20; 95% CI: 0.92–1.57). Additionally, we did not observe significant additive interaction between the dichotomous subjective neighborhood disadvantage index and Alzheimer's disease polygenic score for any cognitive impairment (RERI: −0.02; 95% CI: −0.31–0.27), CIND (RERI: −0.03; 95% CI: −0.32–0.27), or dementia (RERI: 0.01; 95% CI: −0.61–0.63) (S3 Table).

### African ancestries sample

**Multivariable analysis.** We observed that a one standard deviation increase in the subjective neighborhood disadvantage index was associated with a 5% higher risk of any cognitive impairment (95% CI: 0.94–1.17), a 4% higher risk of CIND (95% CI: 0.93–1.17), and a 3% higher risk of dementia (95% CI: 0.90–1.20) in the fully adjusted model; however, these associations were not statistically significant (Table 2). We found no evidence of multiplicative interaction between the subjective neighborhood disadvantage index and Alzheimer's disease polygenic score; however, power to detect small-to-moderate effects was limited.

**RERI analysis.** Similarly, in the RERI models with dichotomized exposures, living in the more disadvantaged neighborhoods was associated with higher risk of developing cognitive impairment, while the association was not significant (S2 Table). In addition, we found no evidence of additive interaction between the subjective neighborhood disadvantage index and Alzheimer's disease polygenic score for any cognitive impairment, CIND, or dementia in the African ancestries sample where power to detect an interaction was limited. (S3 Table).

### Sensitivity analysis

To test the robustness of our models, we added several sensitivity covariates including smoking, alcohol consumption, BMI, diabetes, self-rated hearing, self-rated vision, brain-related conditions, chronic conditions, and depression (S4 Table). After adjustment for sensitivity variables, the associations between subjective neighborhood disadvantage index and Alzheimer's disease polygenic score with the risk of any cognitive impairment and dementia remained robust in the European ancestries sample (S4 Table), however, the association became non-significant in the CIND European ancestries sample (HR: 1.05; 95% CI: 1.00–1.11). Both multiplicative and additive interaction effects between the polygenic score for Alzheimer's disease and subjective neighborhood disadvantage index were non-significant (Table 2 and S3 Table). In addition, all main and interaction effects of subjective neighborhood disadvantage index and Alzheimer's disease polygenic score remained non-significant in the African ancestries sample (Table 2 and S3 Table).

To examine each dimension of neighborhood, we tested each neighborhood factor separately. In the European ancestries samples, when accounting for age, gender, education, poverty, *APOE-ε4*, subjective social status, and genetic principal components in Cox models; neighborhood safety, trust, friendliness, cleanliness, vandalism, and belonging were all individually associated with higher risk of any cognitive impairment, CIND, and dementia in the European ancestries sample (S5 Table). Only neighborhood friendliness was significantly associated with risk of any cognitive impairment and CIND in the African ancestries sample (S5 Table).

## Discussion

Using data from a population-based, longitudinal study of older adults in the United States, we found that living in a more disadvantaged neighborhood was associated with a higher risk of developing incident cognitive impairment in European ancestries. Further, the Alzheimer's disease polygenic score was associated with cognitive impairment in individuals of European ancestries. However, we found no evidence of effect modification between the Alzheimer's disease polygenic score and subjective neighborhood disadvantage index on cognitive impairment, whether evaluated on the multiplicative or additive scale, though power to detect small-to-moderate effects, particularly in the African ancestries, was limited. Our results suggest that neighborhood characteristics influence cognitive impairment risk independent of genetic predisposition - Alzheimer's disease polygenic score. These findings underscore the significant role of neighborhood factors in cognitive impairment and highlight the need for examining targeted interventions in disadvantaged communities to potentially mitigate cognitive decline.

### European ancestries findings and consistency

We observed that perceived neighborhood disadvantage (higher subjective neighborhood disadvantage index) was associated with an increased risk of cognitive impairment in the European ancestries sample. This finding aligns with existing literature. For example, a population-based cross-sectional study of 10,289 middle-aged and older adults indicated that a perceived traffic issue and lack of adequate parks were linked to worse cognitive function [56]. In the National Social Life, Health, and Aging Project, higher levels of perceived neighborhood danger (1 mile radius from the home, or within a 20 minute walk of the home) were associated with lower cognitive function, while residents of more cohesive neighborhoods exhibited better cognitive performance [10]. Similarly, a community-level study conducted in Korea with 1,974,944 participants over 50 years of age found that higher community-level social trust is associated with a reduced risk of dementia [13]. In sensitivity analyses where each perceived

**Table 2.** Hazard Ratios from survival analysis stratified by Ancestry, estimates present the association for each standard deviation increase in the neighborhood disadvantage index with incident cognitive impairment (CIND and dementia), CIND, and dementia, relative to normal cognition and non-dementia respectively in the US Health and Retirement Study (2008-2010 Waves).

| | Cognitive Impairment vs. Normal Cognition, European Ancestry[b] (n=6,123) | | | | | | | | | CIND vs. Normal Cognition, European Ancestry[b] (n=6,051) | | |
|---|---|---|---|---|---|---|---|---|---|---|---|---|
| | Model 1 | | | Model 2 | | | Model 3 | | | Model 1 | | |
| | HR | 95% CI | p-value | HR | 95% CI | p-value | HR | 95% CI | p-value | HR | 95% CI | p-value |
| **Neighborhood** | 1.09 | 1.03, 1.15 | **0.002** | 1.09 | 1.03, 1.15 | **0.002** | 1.09 | 1.03, 1.15 | **0.002** | 1.08 | 1.03, 1.14 | **0.004** |
| **Age** | 1.08 | 1.07, 1.08 | **<0.001** | 1.08 | 1.07, 1.08 | **<0.001** | 1.08 | 1.07, 1.08 | **<0.001** | 1.08 | 1.07, 1.08 | **<0.001** |
| **Sex** | | | | | | | | | | | | |
| Female | Ref | Ref | Ref | Ref | Ref | Ref | Ref | Ref | Ref | Ref | Ref | Ref |
| Male | 1.3 | 1.18, 1.43 | **<0.001** | 1.29 | 1.18, 1.42 | **<0.001** | 1.29 | 1.18, 1.42 | **<0.001** | 1.31 | 1.19, 1.45 | **<0.001** |
| **Education** | | | | | | | | | | | | |
| Above High School/GED | Ref | Ref | Ref | Ref | Ref | Ref | Ref | Ref | Ref | Ref | Ref | Ref |
| High School/GED | 1.62 | 1.44, 1.82 | **<0.001** | 1.62 | 1.44, 1.82 | **<0.001** | 1.62 | 1.44, 1.82 | **<0.001** | 1.65 | 1.46, 1.86 | **<0.001** |
| Less than High School/GED | 3.03 | 2.56, 3.58 | **<0.001** | 3.03 | 2.56, 3.58 | **<0.001** | 3.02 | 2.56, 3.58 | **<0.001** | 3.08 | 2.60, 3.66 | **<0.001** |
| **Poverty Status (Below)** | | | | | | | | | | | | |
| Above Poverty threshold | Ref | Ref | Ref | Ref | Ref | Ref | Ref | Ref | Ref | Ref | Ref | Ref |
| Below Poverty threshold | 1.28 | 0.99, 1.65 | 0.055 | 1.27 | 0.99, 1.64 | 0.062 | 1.27 | 0.99, 1.64 | 0.062 | 1.28 | 0.99, 1.66 | 0.060 |
| **APOE E4 status** | | | | | | | | | | | | |
| No copies of e4 | Ref | Ref | Ref | Ref | Ref | Ref | Ref | Ref | Ref | Ref | Ref | Ref |
| Any copies of e4 | 1.43 | 1.29, 1.59 | **<0.001** | 1.42 | 1.28, 1.57 | **<0.001** | 1.42 | 1.28, 1.57 | **<0.001** | 1.43 | 1.29, 1.60 | **<0.001** |
| **Social Ladder** | 0.93 | 0.90, 0.95 | **<0.001** | 0.93 | 0.90, 0.95 | **<0.001** | 0.93 | 0.90, 0.95 | **<0.001** | 0.92 | 0.89, 0.95 | **<0.001** |
| **Baseline wave** | | | | | | | | | | | | |
| Wave 1 (2008) | Ref | Ref | Ref | Ref | Ref | Ref | Ref | Ref | Ref | Ref | Ref | Ref |
| Wave 2 (2010) | 0.86 | 0.78, 0.94 | **0.001** | 0.86 | 0.79, 0.95 | **0.002** | 0.86 | 0.79, 0.95 | **0.002** | 0.86 | 0.78, 0.95 | **0.002** |
| **PGS-AD** | – | – | – | 1.1 | 1.05, 1.16 | **<0.001** | 1.10 | 1.05, 1.16 | **<0.001** | – | – | – |
| **Neighborhood* PGS-AD** | – | – | – | – | – | – | 0.99 | 0.94, 1.04 | 0.800 | – | – | – |
| RERI: The most disadvantaged neighborhoods*PGS-AD Above 75% [a] | – | – | – | – | – | – | −0.02 | −0.31, 0.27 | | – | – | – |
| | Cognitive Impairment vs. Normal Cognition, African Ancestry[c] (n=703) | | | | | | | | | CIND vs. Normal Cognition, African Ancestry[c] (n=695) | | |
| | Model 1 | | | Model 2 | | | Model 3 | | | Model 1 | | |
| | HR | 95% CI | p-value | HR | 95% CI | p-value | HR | 95% CI | p-value | HR | 95% CI | p-value |
| **Neighborhood** | 1.04 | 0.93, 1.16 | 0.500 | 1.04 | 0.93, 1.16 | 0.500 | 1.05 | 0.94, 1.17 | 0.4 | 1.04 | 0.93, 1.16 | 0.500 |
| **Age** | 1.05 | 1.04, 1.07 | **<0.001** | 1.05 | 1.04, 1.07 | **<0.001** | 1.05 | 1.04, 1.07 | **<0.001** | 1.05 | 1.04, 1.07 | **<0.001** |
| **Sex** | | | | | | | | | | | | |
| Female | Ref | Ref | Ref | Ref | Ref | Ref | Ref | Ref | Ref | Ref | Ref | Ref |
| Male | 1.2 | 0.94, 1.53 | 0.150 | 1.22 | 0.95, 1.56 | 0.110 | 1.22 | 0.96, 1.56 | 0.11 | 1.19 | 0.93, 1.52 | 0.200 |
| **Education** | | | | | | | | | | | | |
| Above High School/GED | Ref | Ref | Ref | Ref | Ref | Ref | Ref | Ref | Ref | Ref | Ref | Ref |
| High School/GED | 1.5 | 1.11, 2.03 | **0.009** | 1.51 | 1.11, 2.04 | **0.009** | 1.51 | 1.11, 2.05 | **0.009** | 1.53 | 1.12, 2.09 | **0.007** |
| Less than High School/GED | 3.02 | 2.12, 4.31 | **<0.001** | 2.94 | 2.06, 4.20 | **<0.001** | 2.97 | 2.08, 4.25 | **<0.001** | 3.11 | 2.17, 4.46 | **<0.001** |
| **Poverty Status** | | | | | | | | | | | | |
| Above Poverty threshold | Ref | Ref | Ref | Ref | Ref | Ref | Ref | Ref | Ref | Ref | Ref | Ref |
| Below Poverty threshold | 1.78 | 1.34, 2.37 | **<0.001** | 1.81 | 1.36, 2.41 | **<0.001** | 1.82 | 1.36, 2.43 | **<0.001** | 1.79 | 1.34, 2.39 | **<0.001** |
| **APOE E4 status** | | | | | | | | | | | | |
| No copies of e4 | Ref | Ref | Ref | Ref | Ref | Ref | Ref | Ref | Ref | Ref | Ref | Ref |

| | Dementia vs. Non-dementia, European Ancestry[b] (n=6,789) | | | | | | | | | | | |
|---|---|---|---|---|---|---|---|---|---|---|---|---|
| **Model 2** | | | **Model 3** | | | **Model 1** | | | **Model 2** | | | **Model 3** | | |
| **HR** | **95% CI** | **p-value** | **HR** | **95% CI** | **p-value** | **HR** | **95% CI** | **p-value** | **HR** | **95% CI** | **p-value** | **HR** | **95% CI** | **p-value** |
| 1.08 | 1.02, 1.14 | **0.004** | 1.08 | 1.02, 1.14 | **0.004** | 1.13 | 1.03, 1.24 | **0.010** | 1.13 | 1.02, 1.24 | **0.014** | 1.13 | 1.02, 1.24 | **0.015** |
| 1.08 | 1.07, 1.08 | **<0.001** | 1.08 | 1.07, 1.08 | **<0.001** | 1.12 | 1.10, 1.13 | **<0.001** | 1.12 | 1.10, 1.13 | **<0.001** | 1.12 | 1.10, 1.13 | **<0.001** |
| | | | | | | | | | | | | | | |
| Ref | Ref | Ref | Ref | Ref | Ref | Ref | Ref | Ref | Ref | Ref | Ref | Ref | Ref | Ref |
| 1.31 | 1.18, 1.44 | **<0.001** | 1.31 | 1.19, 1.44 | **<0.001** | 1.05 | 0.87, 1.28 | 0.600 | 1.06 | 0.87, 1.29 | 0.600 | 1.06 | 0.88, 1.29 | 0.500 |
| | | | | | | | | | | | | | | |
| Ref | Ref | Ref | Ref | Ref | Ref | Ref | Ref | Ref | Ref | Ref | Ref | Ref | Ref | Ref |
| 1.65 | 1.46, 1.86 | **<0.001** | 1.65 | 1.46, 1.86 | **<0.001** | 1.58 | 1.23, 2.04 | **<0.001** | 1.6 | 1.24, 2.06 | **<0.001** | 1.59 | 1.23, 2.06 | **<0.001** |
| 3.08 | 2.60, 3.66 | **<0.001** | 3.08 | 2.59, 3.66 | **<0.001** | 3.93 | 2.92, 5.29 | **<0.001** | 3.98 | 2.95, 5.37 | **<0.001** | 3.98 | 2.95, 5.37 | **<0.001** |
| | | | | | | | | | | | | | | |
| Ref | Ref | Ref | Ref | Ref | Ref | Ref | Ref | Ref | Ref | Ref | Ref | Ref | Ref | Ref |
| 1.27 | 0.98, 1.65 | 0.068 | 1.27 | 0.98, 1.65 | 0.068 | 2.03 | 1.39, 2.98 | **<0.001** | 2.07 | 1.41, 3.04 | **<0.001** | 2.07 | 1.41, 3.04 | **<0.001** |
| | | | | | | | | | | | | | | |
| Ref | Ref | Ref | Ref | Ref | Ref | Ref | Ref | Ref | Ref | Ref | Ref | Ref | Ref | Ref |
| 1.42 | 1.28, 1.58 | **<0.001** | 1.42 | 1.28, 1.58 | **<0.001** | 2.06 | 1.70, 2.50 | **<0.001** | 2.04 | 1.68, 2.48 | **<0.001** | 2.04 | 1.68, 2.48 | **<0.001** |
| 0.92 | 0.90, 0.95 | **<0.001** | 0.92 | 0.89, 0.95 | **<0.001** | 0.98 | 0.92, 1.04 | 0.500 | 0.98 | 0.93, 1.04 | 0.600 | 0.98 | 0.93, 1.04 | 0.6 |
| | | | | | | | | | | | | | | |
| Ref | Ref | Ref | Ref | Ref | Ref | Ref | Ref | Ref | Ref | Ref | Ref | Ref | Ref | Ref |
| 0.87 | 0.79, 0.95 | **0.003** | 0.86 | 0.79, 0.95 | **0.003** | 0.8 | 0.67, 0.97 | **0.022** | 0.8 | 0.67, 0.97 | **0.021** | 0.8 | 0.67, 0.97 | **0.020** |
| 1.10 | 1.05, 1.16 | **<0.001** | 1.10 | 1.05, 1.16 | **<0.001** | – | – | – | 1.05 | 0.96, 1.16 | 0.300 | 1.05 | 0.95, 1.16 | 0.3 |
| – | – | – | 0.99 | 0.94, 1.05 | 0.800 | – | – | – | – | – | – | 0.98 | 0.90, 1.08 | 0.7 |
| – | – | – | −0.03 | −0.32, 0.27 | | – | – | – | – | – | – | 0.01 | −0.61, 0.63 | |

| | Dementia vs. Non-dementia, African Ancestry[c] (n=971) | | | | | | | | | | | |
|---|---|---|---|---|---|---|---|---|---|---|---|---|
| **Model 2** | | | **Model 3** | | | **Model 1** | | | **Model 2** | | | **Model 3** | | |
| **HR** | **95% CI** | **p-value** | **HR** | **95% CI** | **p-value** | **HR** | **95% CI** | **p-value** | **HR** | **95% CI** | **p-value** | **HR** | **95% CI** | **p-value** |
| 1.04 | 0.93, 1.16 | 0.500 | 1.04 | 0.93, 1.17 | 0.500 | 1.02 | 0.89, 1.18 | 0.800 | 1.03 | 0.89, 1.19 | 0.700 | 1.03 | 0.90, 1.20 | 0.6 |
| 1.05 | 1.04, 1.07 | **<0.001** | 1.05 | 1.04, 1.07 | **<0.001** | 1.09 | 1.07, 1.11 | **<0.001** | 1.09 | 1.07, 1.11 | **<0.001** | 1.09 | 1.07, 1.11 | **<0.001** |
| | | | | | | | | | | | | | | |
| Ref | Ref | Ref | Ref | Ref | Ref | Ref | Ref | Ref | Ref | Ref | Ref | Ref | Ref | Ref |
| 1.2 | 0.94, 1.54 | 0.200 | 1.2 | 0.94, 1.54 | 0.15 | 1.32 | 0.96, 1.82 | 0.088 | 1.37 | 0.99, 1.89 | 0.056 | 1.37 | 1.00, 1.89 | 0.053 |
| | | | | | | | | | | | | | | |
| Ref | Ref | Ref | Ref | Ref | Ref | Ref | Ref | Ref | Ref | Ref | Ref | Ref | Ref | Ref |
| 1.54 | 1.12, 2.10 | **0.007** | 1.54 | 1.13, 2.11 | **0.007** | 2.14 | 1.15, 3.96 | **0.016** | 2.21 | 1.19, 4.10 | **0.012** | 2.21 | 1.19, 4.09 | **0.012** |
| 3 | 2.09, 4.30 | **<0.001** | 3.02 | 2.10, 4.34 | **<0.001** | 5.48 | 2.94, 10.2 | **<0.001** | 5.61 | 3.00, 10.5 | **<0.001** | 5.6 | 2.99, 10.5 | **<0.001** |
| | | | | | | | | | | | | | | |
| Ref | Ref | Ref | Ref | Ref | Ref | Ref | Ref | Ref | Ref | Ref | Ref | Ref | Ref | Ref |
| 1.81 | 1.36, 2.42 | **<0.001** | 1.83 | 1.37, 2.44 | **<0.001** | 1.86 | 1.30, 2.65 | **<0.001** | 1.83 | 1.28, 2.61 | **<0.001** | 1.82 | 1.28, 2.61 | **<0.001** |
| | | | | | | | | | | | | | | |
| Ref | Ref | Ref | Ref | Ref | Ref | Ref | Ref | Ref | Ref | Ref | Ref | Ref | Ref | Ref |

*(Continued)*

**Table 2.** (Continued)

| | Cognitive Impairment vs. Normal Cognition, European Ancestry[b] (n = 6,123) | | | | | | | | | CIND vs. Normal Cognition, European Ancestry[b] (n=6,051) | | |
|---|---|---|---|---|---|---|---|---|---|---|---|---|
| | Model 1 | | | Model 2 | | | Model 3 | | | Model 1 | | |
| | HR | 95% CI | p-value | HR | 95% CI | p-value | HR | 95% CI | p-value | HR | 95% CI | p-value |
| Any copies of e4 | 1.01 | 0.80, 1.28 | >0.9 | 1.02 | 0.80, 1.29 | 0.900 | 1 | 0.79, 1.28 | >0.9 | 1 | 0.79, 1.27 | >0.9 |
| **Social Ladder** | 0.97 | 0.91, 1.03 | 0.300 | 0.96 | 0.90, 1.02 | 0.200 | 0.96 | 0.90, 1.02 | 0.2 | 0.97 | 0.91, 1.04 | 0.400 |
| **Baseline wave** | | | | | | | | | | | | |
| Wave 1 (2008) | Ref | Ref | Ref | Ref | Ref | Ref | Ref | Ref | Ref | Ref | Ref | Ref |
| Wave 2 (2010) | 1.19 | 0.94, 1.51 | 0.150 | 1.17 | 0.92, 1.48 | 0.200 | 1.17 | 0.92, 1.48 | 0.2 | 1.2 | 0.94, 1.52 | 0.150 |
| **PGS-AD** | – | – | – | 1.13 | 0.95, 1.35 | 0.200 | 1.15 | 0.96, 1.38 | 0.12 | – | – | – |
| **Neighborhood* PGS-AD** | – | – | – | – | – | – | 0.94 | 0.82, 1.07 | 0.3 | – | – | – |
| RERI: The most disadvantaged neighborhoods *PGS-AD Above 75% [a] | – | – | – | – | – | – | −0.42 | −1.09, 0.25 | | – | – | – |

[a]RERI was calculated based on the model using a dichotomized neighborhood disadvantage index (≤ 0 indicating the least disadvantaged neighborhoods; > 0 indicating the most disadvantaged neighborhoods) and PGS-AD (cutoff at the 75th percentile). Full model details are provided in S3 Table.

[b]European Ancestry Sample average follow-up duration: Cognitive Impairment = 7.49 years; Cognitive Impairment, No Dementia (CIND) = 7.53 years; Dementia = 8.04 years.

[c]African Ancestry Sample average follow-up duration: Cognitive Impairment = 6.71 years; Cognitive Impairment, No Dementia (CIND) = 6.77 years; Dementia = 7.85 years.

neighborhood characteristic was considered individually, neighborhood safety, cleanliness, trust, cleanliness, vandalism and belonging were significantly associated with the risk of cognitive impairment in the European ancestries sample. This indicates that both perceived neighborhood environment and neighborhood connections are crucial risk factors for cognitive health.

### African ancestries findings and consistency

The association between subjective neighborhood disadvantage index and cognitive impairment was also observed in African ancestries participants, but the estimate may be imprecise, with a 95% confidence interval crossing the null. We did note a significant association between higher perceived neighborhood friendliness and the lower risk of cognitive impairment in the African ancestries participants. Existing studies indicate that, despite the significant growth of the middle-class Black Americans in the United States, many continue to live in inferior neighborhoods compared to white counterparts [57–59]. This suggests that even with rising income levels, African ancestry communities still encounter structural inequalities that limit their access to better housing and living environments. Furthermore, these inequalities extend beyond housing to areas like education, employment, and healthcare, affecting the overall health of minority populations [60]. Neighborhood friendliness and social support, while not the only factor, could play a critical role in mitigating the stressors that contribute to cognitive impairment in African ancestry population [61]. Additionally, the nature of the HRS may introduce selective survival bias, as those experiencing a higher disease burden are less likely to be included or remain in the study [62], potentially underestimating the impact of disadvantaged neighborhoods on cognitive impairment, biasing results towards the null. This issue maybe more pronounced among racial minority groups, who are disproportionately affected by cumulative disadvantage, chronic disease, and premature mortality compared to Non-Hispanic White populations [63,64].

### Neighborhood characteristics

We utilized a subjective neighborhood disadvantage index derived through principal component analysis to present neighborhood characteristics. While this approach reduces dimensionality by integrating correlated variables into a

| | | | | | | Dementia vs. Non-dementia, European Ancestry[b] (n = 6,789) | | | | | | | | |
|---|---|---|---|---|---|---|---|---|---|---|---|---|---|---|
| Model 2 | | | Model 3 | | | Model 1 | | | Model 2 | | | Model 3 | | |
| HR | 95% CI | p-value | HR | 95% CI | p-value | HR | 95% CI | p-value | HR | 95% CI | p-value | HR | 95% CI | p-value |
| 1.01 | 0.79, 1.28 | >0.9 | 0.99 | 0.78, 1.26 | >0.9 | 1.44 | 1.05, 1.96 | **0.024** | 1.46 | 1.06, 2.00 | **0.020** | 1.43 | 1.04, 1.97 | **0.026** |
| 0.96 | 0.90, 1.03 | 0.300 | 0.96 | 0.90, 1.03 | 0.3 | 1.06 | 0.98, 1.16 | 0.200 | 1.06 | 0.97, 1.15 | 0.200 | 1.06 | 0.97, 1.15 | 0.2 |
| | | | | | | | | | | | | | | |
| Ref | Ref | Ref | Ref | Ref | Ref | Ref | Ref | Ref | Ref | Ref | Ref | Ref | Ref | Ref |
| 1.17 | 0.92, 1.50 | 0.200 | 1.17 | 0.92, 1.50 | 0.2 | 1.12 | 0.82, 1.53 | 0.500 | 1.1 | 0.80, 1.50 | 0.600 | 1.11 | 0.81, 1.52 | 0.5 |
| 1.11 | 0.93, 1.33 | 0.200 | 1.13 | 0.95, 1.35 | 0.2 | – | – | – | 1.09 | 0.86, 1.39 | 0.500 | 1.12 | 0.88, 1.44 | 0.4 |
| – | – | – | 0.94 | 0.82, 1.08 | 0.4 | – | – | – | – | – | – | 0.92 | 0.78, 1.09 | 0.3 |
| – | – | – | −0.33 | −0.97, 0.31 | | – | – | – | – | – | – | −0.14 | −0.96, 0.68 | |

composite measure, it may not fully capture the variability across neighborhood characteristics. To address this limitation, we conducted sensitivity analyses by examining each neighborhood characteristic individually, providing a more nuanced perspective. Additionally, the index included only seven dimensions of subjective neighborhood characteristics, which may not comprehensively reflect the range of relevant factors. Future research should incorporate additional variables, such as walkability, aesthetic appeal, and neighborhood diversity, to achieve a more complete assessment. Furthermore, we measured neighborhood factors and covariates only at baseline. Although neighborhood characteristics tend to be relatively stable – especially in this sample, the results would be more precise and dynamic if neighborhood factors were included as time-varying variables in future analyses. Our focus on subjective neighborhood characteristics is a strength, as these measures capture direct mental and emotional responses to the environment. Nevertheless, future research should also incorporate objective neighborhood measures to complement and validate our findings.

### Genetic factors

We found the association between the Alzheimer's disease polygenic score and cognitive impairment only among the European ancestries participants. Although previous studies have identified significant associations between the Alzheimer's disease polygenic score and cognitive impairment in both European and African ancestries populations [65–67], some findings suggest that these associations may depend on the inclusion of the *APOE* locus [68–71]. We observed a significant association between *APOE*-ε4 status and cognitive impairment in both European and African ancestries samples. This is consistent with another study, which found that participants with any APOE-ε4 allele had 2.42 times higher odds of developing dementia in European ancestry samples and 1.77 times higher odds in African ancestry samples compared to those without the allele [72]. The robust association of *APOE*-ε4 with cognitive impairment across diverse populations reinforces its role as a significant genetic risk factor for Alzheimer's disease and related cognitive declines [73]. Moreover, while the limited sample size of African ancestry participants may reduce the statistical power to detect significant associations, additional factors related to ancestry-specific polygenic risk scores also warrant consideration. Specifically, the polygenic risk scores used in this analysis were derived from GWASs primarily conducted in European ancestry populations [74]. Such GWASs often have limited transferability to non-European populations due to differences in linkage disequilibrium, minor allele frequency, heritability, and genetic correlation across ancestries [74]. Additionally, the included analytic European ancestries sample had a lower Alzheimer's disease polygenic score than the excluded sample due to missingness of covariates. This pattern raises the possibility of selection bias, which could potentially lead to underestimation of the true association between the Alzheimer's disease polygenic score and cognitive impairment in the European ancestries.

### Strengths

Our study stratified by ancestries, allowing for the examination of interactions between genetic and neighborhood factors within specific population subgroups. Secondly, we employed longitudinal cognitive assessments to evaluate the risk of developing cognitive impairment over time which is a strength compared to cross-sectional studies. Additionally, we incorporated subjective measurements of neighborhood factors across seven dimensions: vandalism, cleanliness, vacancies, trust, belonging, friendliness, and safety. This detailed exploration enables a nuanced understanding of how various neighborhood aspects contribute to cognitive health outcomes over time.

### Conclusions

Our study investigated the relationship between seven subjective neighborhood characteristics—both collectively and individually—and genetic factors on the risk of cognitive impairment. We found that living in a more disadvantaged neighborhood and having a higher genetic risk for Alzheimer's disease were significantly associated with an increased risk of developing any cognitive impairment, CIND, and dementia among European ancestries participants. However, the association of neighborhood characteristics on cognitive impairment risk did not vary based on genetic risk for Alzheimer's disease. Given the evidence provided from this study and previous literature, we suggest that neighborhood is an important modifiable risk factor for cognitive health. Public health interventions should prioritize improving neighborhood safety, cleanliness, and fostering community connections (trust, friendliness and belonging), as these factors are significantly associated with the risk of cognitive impairment. Future research should incorporate longitudinal assessments of neighborhood characteristics to enhance the robustness of findings.

### Supporting information

**S1 Table. Characteristics of included and excluded participants with neighborhood disadvantage index in the Health and Retirement Study (2008–2010 Waves).** [a]n (%); Mean (SD). [b]Pearson's Chi-squared test; Welch Two Sample t-test.
(DOCX)

**S2 Table. Sample distribution by genetic ancestry in the Health and Retirement Study, wave 2008–2010.** [a]n (%); Mean (SD).
(DOCX)

**S3 Table. Hazard ratios from survival analysis with binary neighborhood disadvantage index stratified by ancestry, estimates present the association for the most disadvantaged neighborhoods with incident cognitive impairment (CIND and dementia), CIND and dementia, relative to normal cognition and non-dementia respectively in the US Health and Retirement Study (2008–2010 Waves).**
(DOCX)

**S4 Table. Hazard ratios from survival analysis with additional sensitivity covariates stratified by ancestry, estimates present the association for each standard deviation increase in the neighborhood disadvantage index with incident cognitive impairment (CIND and dementia), CIND and dementia, relative to normal cognition and non-dementia respectively in the US Health and Retirement Study (2008–2010 Waves).** Each model further adjusted for Smoking, Alcohol Consumption, BMI, Diabetes, Self-Rated Hearing and Vision, Brain-related Conditions, Chronic Conditions, and Depression.
(DOCX)

**S5 Table. Hazard ratios from survival analysis using individual neighborhood factor as exposure, stratified by ancestry: risk of incident cognitive impairment (CIND and dementia), CIND and dementia, relative to normal**

**cognition and non-dementia in the US Health and Retirement Study (2008–2010 Waves).** All models adjusted for Age, Sex, Education, Poverty Status, APOE E4 Status, Social Ladder, and Baseline Wave.
(DOCX)

**S6 Table. Hazard ratios from survival analysis comparing binary neighborhood disadvantage index with additional covariates, stratified by ancestry: risk of incident cognitive impairment (CIND and dementia), CIND and dementia in the US Health and Retirement Study (2008–2010 Waves).**
(DOCX)

## Acknowledgments

We thank the participants and staff of the Health and Retirement Study.

## Author contributions

**Conceptualization:** Erin B. Ware, Kelly M. Bakulski.

**Data curation:** Erin B. Ware, Peiyao Zhu, Mikayla Benbow, Lindsay H. Ryan, Kelly M. Bakulski.

**Formal analysis:** Peiyao Zhu, Mikayla Benbow.

**Funding acquisition:** Erin B. Ware, Lindsay H. Ryan, Kelly M. Bakulski.

**Methodology:** Erin B. Ware, Peiyao Zhu, Grace Noppert, Mikayla Benbow, Lindsay C. Kobayashi, Lindsay H. Ryan, Kelly M. Bakulski.

**Project administration:** Erin B. Ware, Kelly M. Bakulski.

**Resources:** Erin B. Ware, Lindsay C. Kobayashi, Kelly M. Bakulski.

**Supervision:** Erin B. Ware, Lindsay C. Kobayashi, Lindsay H. Ryan, Kelly M. Bakulski.

**Validation:** Peiyao Zhu, Mingzhou Fu.

**Writing – original draft:** Erin B. Ware, Peiyao Zhu, Mikayla Benbow, Kelly M. Bakulski.

**Writing – review & editing:** Erin B. Ware, Peiyao Zhu, Grace Noppert, Mingzhou Fu, Mikayla Benbow, Lindsay C. Kobayashi, Lindsay H. Ryan, Kelly M. Bakulski.

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
