## [Decision Letter · Decision Letter 0]

29 Sep 2025

Dear Dr. Ware,

Thank you for submitting your manuscript to PLOS ONE. After careful consideration, we feel that it has merit but does not fully meet PLOS ONE’s publication criteria as it currently stands. Therefore, we invite you to submit a revised version of the manuscript that addresses the points raised during the review process.

We look forward to receiving your revised manuscript.

Kind regards,

Mu-Hong Chen, M.D., Ph.D.

Academic Editor

PLOS ONE

Journal Requirements:

Reviewers' comments:

Reviewer's Responses to Questions

**Comments to the Author**

1. Is the manuscript technically sound, and do the data support the conclusions?

Reviewer #1: Yes

Reviewer #2: Yes

2. Has the statistical analysis been performed appropriately and rigorously?

Reviewer #1: Yes

Reviewer #2: Yes

3. Have the authors made all data underlying the findings in their manuscript fully available?

Reviewer #1: Yes

Reviewer #2: Yes

4. Is the manuscript presented in an intelligible fashion and written in standard English?

Reviewer #1: Yes

Reviewer #2: Yes

Reviewer #1: This is a rigorous and creative analytic approach. It not only focuses on the primary research topic (neighborhood environment) but also simultaneously takes dementia polygenic risk into account.

I only have a few questions to raise:

In the covariates included for the sensitivity analysis, according to The Lancet 2024 dementia risk factor report, the following are missing: hearing impairment, visual impairment, traumatic brain injury, and air pollution.

(I understand there may be design limitations, especially in measuring air pollution, but please at least acknowledge this in the limitations section.)

In addition to the above, I personally believe that substance use and epilepsy should also be considered.

Regarding the categorization of chronic conditions (0, 1–2, ≥3), I think this approach is debatable. I recommend that brain-related conditions (psychiatric disorders, stroke) be separated out, while the remaining physical comorbidities could either be assessed using the CCI score or treated as individual covariates.

Reviewer #2: This cohort study in HRS (2008–2020) examines whether perceived neighborhood disadvantage and Alzheimer’s disease polygenic score (APOE-excluded) predict incident cognitive impairment. In European-ancestry participants, both exposures independently associate with higher risk; no multiplicative or additive interaction is detected. African-ancestry estimates are imprecise. I only have a few comments on this article.

Abstract

1. In the Results, please avoid vague phrases such as “similar effect sizes” and “comparable but nonsignificant trends were noted in the African-ancestry sample”; instead, report the numerical estimates (e.g., HRs with 95% CIs).

Results

2. Given the small African ancestries sample and limited power, the phrasing “comparable but nonsignificant trends” should be tempered to “evidence was inconclusive in African ancestries due to limited precision.”

Discussion

3. Replace any phrasing implying the absence of interaction with language that emphasizes evidence not detected and limited precision, especially for African ancestries: e.g., “We found no evidence of multiplicative or additive interaction; however, power to detect small-to-moderate effects—particularly in African ancestries—was limited.”

4. Avoid “comparable but nonsignificant trends.” Use: “Estimates in African ancestries were directionally similar but imprecise, and the evidence was inconclusive due to limited precision and cross-ancestry PRS transferability.”

**Do you want your identity to be public for this peer review?** For information about this choice, including consent withdrawal, please see our Privacy Policy

Reviewer #1: No

Reviewer #2: No

---

## [Author Response · Author response to Decision Letter 1]

20 Oct 2025

See attached document "Response to Reviewers"

---

## [Decision Letter · Decision Letter 1]

26 Oct 2025

Associations of Perceived Neighborhood Factors and Alzheimer’s Disease Polygenic Score with Cognition: Evidence from the Health and Retirement Study

PONE-D-25-46504R1

Dear Dr. Erin B Ware,

We’re pleased to inform you that your manuscript has been judged scientifically suitable for publication and will be formally accepted for publication once it meets all outstanding technical requirements.

Kind regards,

Mu-Hong Chen, M.D., Ph.D.

Academic Editor

PLOS ONE

Additional Editor Comments (optional):

Reviewers' comments:

Reviewer's Responses to Questions

**Comments to the Author**

Reviewer #1: All comments have been addressed

Reviewer #2: All comments have been addressed

2. Is the manuscript technically sound, and do the data support the conclusions?

Reviewer #1: Yes

Reviewer #2: Yes

3. Has the statistical analysis been performed appropriately and rigorously?

Reviewer #1: Yes

Reviewer #2: Yes

4. Have the authors made all data underlying the findings in their manuscript fully available?

Reviewer #1: Yes

Reviewer #2: Yes

5. Is the manuscript presented in an intelligible fashion and written in standard English?

Reviewer #1: Yes

Reviewer #2: Yes

Reviewer #1: The author did a good job. All comment had been addressed. I have no further comment. ..............

Reviewer #2: The authors have fully addressed all the questions I raised, and I have no further comments or additional feedback at this time.

**Do you want your identity to be public for this peer review?** For information about this choice, including consent withdrawal, please see our Privacy Policy

Reviewer #1: No

Reviewer #2: No

---

## [Editor Report · Acceptance letter]

PONE-D-25-46504R1

PLOS ONE

Dear Dr. Ware,

I'm pleased to inform you that your manuscript has been deemed suitable for publication in PLOS ONE. Congratulations! Your manuscript is now being handed over to our production team.

Kind regards,

on behalf of

Dr. Mu-Hong Chen

Academic Editor

PLOS ONE